# Effectiveness of a Mobile App in Reducing Therapeutic Turnaround Time and Facilitating Communication between Caregivers in a Pediatric Emergency Department: A Randomized Controlled Pilot Trial

**DOI:** 10.3390/jpm12030428

**Published:** 2022-03-09

**Authors:** Frederic Ehrler, Carlotta Tuor, Remy Trompier, Antoine Berger, Michael Ramusi, Robin Rey, Johan N. Siebert

**Affiliations:** 1Division of Medical Information Sciences, Geneva University Hospitals, 1205 Geneva, Switzerland; frederic.ehrler@hcuge.ch; 2Faculty of Medicine, University of Geneva, 1205 Geneva, Switzerland; carlotta.tuor@etu.unige.ch (C.T.); robin.rey@etu.unige.ch (R.R.); 3Information Technology Department, Geneva University Hospitals, 1205 Geneva, Switzerland; remy.trompier@hcuge.ch (R.T.); antoine.berger@hcuge.ch (A.B.); michael.ramusi@hcuge.ch (M.R.); 4Department of Pediatric Emergency Medicine, Children’s Hospital, Geneva University Hospitals, 1205 Geneva, Switzerland

**Keywords:** mobile application, mHealth, digital technology, emergency service, hospital, emergency department, clinical laboratory information systems, communication, text messaging, pediatrics

## Abstract

For maintaining collaboration and coordination among emergency department (ED) caregivers, it is essential to effectively share patient-centered information. Indirect activities on patients, such as searching for laboratory results and sharing information with scattered colleagues, waste resources to the detriment of patients and staff. Therefore, we conducted a pilot study to evaluate the initial efficacy of a mobile app to facilitate rapid mobile access to central laboratory results and remote interprofessional communication. A total of 10 ED residents and registered nurses were randomized regarding the use of the app versus conventional methods during semi-simulated scenarios in a pediatric ED (PED). The primary outcome was the elapsed time in minutes in each group from the availability of laboratory results to their consideration by participants. The secondary outcome was the elapsed time to find a colleague upon request. Time to consider laboratory results was significantly reduced from 23 min (IQR 10.5–49.0) to 1 min (IQR 0–5.0) with the use of the app compared to conventional methods (92.2% reduction in mean times, *p* = 0.0079). Time to find a colleague was reduced from 24 min to 1 min (i.e., 93.0% reduction). Dedicated mobile apps have the potential to improve information sharing and remote communication in emergency care.

## 1. Introduction

Emergency department (ED) overcrowding is a global healthcare problem [1] that is both a source and consequence of prolonged ED length of stay (ED-LOS). Prolonged ED-LOS negatively impacts patients’ waiting times by reducing the efficiency and quality of care, patient satisfaction, and willingness to return [2,3]. For a long time, it has also been recognized as a critical threat to patient safety [4]. ED-LOS is the most compelling and impactful indicator of the level of overcrowding and the organizational capacity of EDs to address this issue [5]. The entire laboratory testing process, also known as therapeutic turnaround time (TAT), is a major contributor [6,7]. It can be schematically divided into three phases (pre-, intra-, and postanalytical) that extend from the physician’s request for testing to the awareness of the results (Appendix A). There is no doubt that shortening the analytical phase can, in principle, shorten ED-LOS by allowing earlier decision making for the patient. However, it is estimated that up to 96% of the delays that contribute most to the therapeutic TAT occur in the pre- and post-analytical phases outside of the central laboratory [8,9]. In particular, caregivers’ delayed awareness and review of results in the post-analytical phase have been described as the largest component of perceived TAT [10]. This makes them a prime target for interventions to mitigate their impact on ED-LOS and improve patient flow. This delay may be partially due to the fact that, while waiting for results without the support of individualized real-time laboratory prompts, caregivers are extremely busy, juggling multiple other responsibilities and multitasking, which take them away from fixed, computerized workstations and from the time the results are released. This places a cognitive burden on caregivers, who must regularly inquire about the availability of laboratory results with the risk of forgetting them. This also requires them to make incessant, time-consuming, and distracting trips to verify their availability.

Emergency care is very complex. It requires patient-centered care coordinated among multiple busy providers in a highly unpredictable and stressful environment, requiring high functional operability. Coupled with the role played by the built environment and the functional need to care for multiple patients simultaneously, this contributes to scatter caregivers, who must work as a team and in close vicinity to ensure quality care and a timely and seamless process. It also prevents accurate, synchronous communication [11] and safe collaborative emergency care [12,13]. Up to 80% of healthcare professionals’ time in the ED is spent on communication in addition to any other tasks actively being performed at the same time, such as medication handling [14]. Approximately 90% of information transactions involve informal interpersonal exchanges rather than interaction with formal information sources, with 82% of synchronous face-to-face communication [14,15]. Disruption of communication in the ED therefore makes it difficult for caregivers to maintain a high level of awareness of each patient’s individual situation; to follow up quickly on colleagues’ requests for support, laboratory tests, radiologic examinations; or to respond to patients’ demands. It has been observed that ED physicians and nurses waste nearly half of their time on indirect patient-related and non-patient-related activities. This includes traveling within the unit and locating colleagues to enable face-to-face communication and to share medical information [16,17,18,19]. Peters et al. showed that ED resident physicians walked an average 4.2 km and attending physicians 3.9 km during a typical 8.5 h shift [20]. This has a direct impact on patients, who must then wait unnecessarily in the ED when they could have been sent home or hospitalized earlier, which would have potentially reduced their LOS. Wasted time, discontinuity of care, suboptimal communication, and the omission or delay in seeking information can compromise patient outcomes and safety.

To our knowledge, no app that can be used as an interoperable mobile laboratory results viewer in the ED has been described to date [21]. Evidence demonstrating the impact of mobile apps tailored to caregivers and specifically dedicated to streamlining shared patient management in the EDs remains scarce [22,23,24,25]. In a previous study [26], we described the user-centered development and high early technology acceptance of a mobile app—the ‘’Patients In My Pocket in my Hospital’’ (PIMPmyHospital) app. The app is designed to help ED caregivers automatically obtain relevant information about the patients they care for in real time, including laboratory and imaging results. The app also provides an end-to-end encrypted chat and instant messaging platform to digitally and remotely connect physicians and nurses caring for the same patients. However, the potential effectiveness of this app in mitigating ED-LOS by obtaining laboratory results in a shorter time frame than by conventional methods as well as its ability to facilitate communication in a shorter time frame among caregivers remained to be evaluated.

## 2. Materials and Methods

### 2.1. Study Design and Participants

This prospective, single-center, non-blind, two-arm, randomized, controlled pilot trial was conducted on 6 September 2021, at the pediatric ED (PED) of the Children’s Hospital in Geneva, which is part of the Geneva University Hospitals’ network. It is one of the largest tertiary PED in Switzerland, with a total of approximately 33,000 visits per year for a resident population of more than 508,000 individuals, of whom 17.5% are children under 16 years of age. The present study was approved by the Geneva Cantonal Ethics Committee/SwissEthics (Req-2021-00795: date of approval, 23 July 2021) and registered at ClinicalTrials.gov (accessed on 13 February 2022) (NCT05203146, principal investigator: JNS, registration date: 24 January 2022). The trial was carried out in accordance with the principles of the Declaration of Helsinki and Good Clinical Practice guidelines [27,28] and adheres to the applicable CONSORT guidelines [29]. Written informed consent was obtained from each participant after full information disclosure prior to study participation.

We evaluated two methods for considering laboratory results (i.e., the post-analytical phase) and finding a colleague to aim for joint action during standardized, semi-simulated scenarios of everyday life in a PED. Participants were randomly assigned to undertake these two actions either with the support of the app (intervention group) or by conventional methods (control group). We hypothesized that use of the app could reduce both the time to learn about laboratory results and the time to interact with colleagues through remote communication.

Eligible participants were postgraduate residents pursuing a <6-year residency in pediatrics and registered nurses from the PED (aged > 18 years). They should have previously attended a standardized 5 min introductory course on the use of the app dispensed by the study investigators on the day of participation. Participants who had not taken the introductory course were excluded. According to Cohen’s calculation, participants were recruited based on an expected effect size of 2.0, a type I error of 0.05, and a power of 80%.

### 2.2. Randomization and Blinding

Participants were randomized using a single, constant 1:1 allocation ratio determined by means of web-based software [30]. Allocation concealment was ensured with the allocation software and was not released until participants started the scenario. Participants were unaware of the scenario during recruitment to minimize preparation bias. After randomization at the beginning of the scenario, they were unblinded to the study arm. The study members (C.T., R.R., and J.N.S.) involved in the scenario and playing the assumed role of a fellow nurse to be found by participants were revealed to participants before the scenario started. Although the intervention could not be masked, all investigators remained unaware of the outcomes until all data were unlocked for analysis at the end of the trial.

### 2.3. The PIMPmyHospital App

The app was developed in an iterative way following a user-centered approach to meet end-users’ needs. In a first phase, semi-structured interviews were conducted with nurses and physicians of the PED to identify their main complaints related to their daily practice. Responses were then structured and used to identify the main functionalities that a mobile app should have in order to be likely to address these grievances, namely (1) an overview of patients currently being cared for in the ED; (2) the possibility to communicate between caregivers through a secure instant messaging system; and (3) notifications when laboratory results are available. The system was implemented using a client-server architecture. The backend is composed of four microservices developed on Java Spring Boot framework. Each microservice is responsible for delivering a specific piece of information to the frontend, i.e., patient information, laboratory results, and instant messaging, and the last one is in charge of notifications. The instant messaging chat server is based on the Rocket.Chat platform installed locally on the hospital’s digital infrastructure. The microservices exchange information with the frontend through a restful application programming interface. The frontend is a hybrid application developed in the Angular and Ionic frameworks. Security is enhanced by using JSON Web tokens obtained during caregiver authentication via a Keycloack identity and access management solution. Authentication on the device relies on two factors, including a login password and a specific certificate installed on the user’s device. Moreover, patient medical data does not persist outside the primary system containing the source data (i.e., the patient electronic health record (EHR) located on the hospital’s secure server). In other words, no data persistence is stored on the app itself.

The app runs on Google’s Android and Apple’s iOS operating systems. Its design and visual aspect have already been published [26]. In brief, a start screen allows caregivers to have an automatically generated view centered on the patients under their care; hence, the origin of the app is the “Patients In My Pocket” solution, referring to phone pocket carriage (Figure 1). Next to each patient, there is a clickable list of other caregivers in charge of the same patient, with whom they can be in contact remotely and collaboratively through the app via a secure instant messaging system. To make it easier to read, a color code is specifically assigned to each profession. Push notifications inform caregivers of incoming messages and whether they have been taken into account to ensure the responsiveness of the messaging system. Colored bars represent the triage level of each patient according to the five degrees of the Canadian triage and acuity scale [31]. The gender and identity of the patients are shown as well as the time elapsed since admission represented as circular radial timers. Furthermore, the geospatial localization of the patient in the department and the status of the patient (e.g., seen by a physician, waiting for results, waiting for a radiological examination, waiting for treatment, etc.) are displayed. Push notifications encourage the user to be attentive to the evolution of situations in real time. Selecting a patient on this page opens a new one containing drop-down contextual tabs with information related to laboratory results, imaging, electrocardiogram results, etc., extracted from the hospital’s patient EHR. Again, push notifications indicate the availability of results. Thus, the entire patient management process is available to the caregiver in the palm of the hand and can be consulted at any time on the move.

### 2.4. Intervention

On the day of participation after randomized allocation, each participant completed a survey to collect data regarding his/her demographic characteristics. Then, they attended a standardized 5 min training session on how to use the mobile app (thus providing identical preliminary education). At this stage, this introductory course was not intended to test the usability of the app but to explain its use for the upcoming intervention. For two reasons, signage and layout of the PED were not modified for the purposes of the study. On the one hand, it is the daily working environment of the participants, who were theoretically supposed to have a similar knowledge of it after a common minimum assignment to the PED of at least 3 months. On the other hand, the study aimed at studying the impact of the app in a real PED environment during the overload period related to the coronavirus disease 2019 (COVID-19) pandemic.

Each participant was then exposed to the semi-simulated scenario about a fictitious patient mixed with the actual clinical activities of the day. They were first required to retrieve the simulated patient’s laboratory results from the EHR when randomly made available. Only the indication at the beginning of the scenario that a laboratory test had just been sent to the central laboratory was provided to participants. They were then free to check the availability of the results either on the app or by using the institutional computerized system (i.e., without app support) at intervals of their choice. Once the result was obtained, participants were then asked to find a nurse (played by a study investigator: C.T., R.R., or J.N.S.) on the ward. This was done either at the prompt of a text message appearing on the app or by an oral request made by a second study investigator. Since one of the main communication problems in EDs is related to the distance between caregivers, this task aimed at determining whether remote communication via the app could generate a common and timely goal-oriented response. No information on the location of the nurse was provided. Participants were free to look for the nurse through a communication with the app for those who had it or, in both cases, to walk around to find the nurse. In all cases, participants ultimately had to physically reach the nurse and could not just make virtual contact. Procedures were standardized to follow the same chronological progression in order to ensure that each participant was exposed to exactly the same case with similar challenges in app usage and decision making. We did not organize pre-tests to minimize preparation bias so as not to influence the intervention. The app was interfaced on an Apple iPhone X with the latest version of iOS, but the app works identically on Android OS (Mountain View, CA, USA). Only the study investigators had access to the data.

### 2.5. Measurement Instruments

Time-to-goal completion was measured similarly between participants using a stopwatch. Given the pilot nature of the study, access logs on the app were not measured at this stage, similar to distances traveled using a pedometer and radio frequency identification (RFID) sensors, which will be the subject of outcomes in a subsequent larger trial. Data collection was carried out using the latest version of Prism 9 for MacOS (GraphPad software, LLC., San Diego, CA, USA).

### 2.6. Outcomes

The primary outcome was the elapsed time (in minutes) in each allocation group from the availability of the new laboratory results on either the mobile app or the institutional computerized patient data system to their consideration by the participant on the allocated medium (i.e., mobile app or EHR). The upper bound was arbitrarily set at 120 min. The secondary outcome was the elapsed time (in minutes) from the moment the participant was informed by the mobile app or a statement given by a study investigator (conventional method) that a nurse required assistance to perform a technical procedure up to the point in time when the participant reached that nurse. The upper bound was again arbitrarily set at 120 min.

### 2.7. Statistical Analysis

For the primary outcome, we first evaluated the time elapsed between the availability of the laboratory results and consideration by the participants. The Shapiro–Wilks test was used for normality analysis of the parameters. As continuous variables were non-normally distributed, the Mann–Whitney test to compare independent groups was used. No paired data were compared. Kaplan–Meier curves were estimated and compared using the log-rank (Mantel–Cox) test for bivariate survival analysis. For the secondary outcome, the same analyses were done. Statistical tests were two-tailed with a significance level of 5%. Data analysis was carried out using Prism v9.3.1 (GraphPad software, LLC., San Diego, CA, USA) for MacOS.

## 3. Results

### 3.1. Overview

A total of 10 participants (5 residents and 5 nurses) completed the scenarios, without dropouts.

Figure 2 presents the CONSORT flowchart for the present randomized, controlled trial. Table 1 summarizes participants’ demographic and healthcare characteristics.

### 3.2. Primary Outcome

Figure 3 shows that the median time to review laboratory results once available was significantly reduced from 23 min (interquartile range (IQR) 10.5–49.0 min) to 1 min (IQR 0–5.0 min) with the use of the app compared to the patient’s EHR. All participants who used the app accessed these results at most within 7 min of their publication, whereas it took up to 58 min for participants who did not use the app (Figure 4), i.e., a 92.2% reduction in mean post-analytical time to consider the laboratory results.

### 3.3. Secondary Outcome

Median time to find a colleague was reduced almost significantly from 24 min to 1 min with the use of the app compared to the patient’s EHR (Figure 3). All participants who used the app found their colleague at most within 4 min, whereas it took up to 56 min for participants who did not use the app (Figure 4), i.e., a 93.0% reduction in time wasted searching for someone.

## 4. Discussion

In this randomized pilot trial, we found that the use of a user-centered mobile app significantly reduced the time taken by pediatric emergency caregivers to consider laboratory results as well as the time needed to find colleagues. The result was borderline significant for the latter, however, likely related to the small sample size. The results of this pilot study will be used to inform the design of a future larger-scale, time-motion, randomized, controlled trial to confirm the effectiveness of PIMPmyHospital in reducing time wasted and the distance traveled to find laboratory results and colleagues.

According to the U.S. Centers for Disease Control and Prevention [32], laboratory tests are ordered in nearly 48% of ED visits, and 70% of medical decisions today are based on these [33], thus showing their important contribution in emergency care. Lack of timely follow-up of laboratory results has been widely documented as contributing to prolonged hospital LOS [34]. Their analytic phase being intrinsically linked to the incompressible speed of the measuring instruments used as well as the technological and organizational progress made in the field of centralized laboratory testing offers a limited potential for further shortening this phase [35]. This is partly at odds with the concerns of emergency physicians who are willing to sacrifice some analytical quality for a faster TAT [9]. This can explain the craze for point-of-care testing (POCT) as an alternative to provide timely results, as it speeds up clinical decision making and relieves ED congestion by expediting patient flow [36,37]. However, there is still controversy about its ability to reduce the total ED-LOS [38]. Of note, POCT does not a priori eliminate the risk that caregivers’ awareness and consideration of the results will be similarly delayed, as this is a post-analytical issue. Prescribing POCT earlier in management would not necessarily change this situation and even have the unintended consequence of increasing the total number of tests [39]. Furthermore, POCTs are often more specialized and limited in their overall function compared to more advanced tests that continue to be performed in the central laboratory [40].

Our preliminary results show that in addition to being well accepted by caregivers [26], an app, such as PIMPmyHospital, could fulfill this role. Its mobile nature, supported by push notifications on their smartphones, should allow caregivers to avoid sacrificing their time to seek laboratory results but to obtain them immediately upon release wherever they are at the point of care. It has been suggested that the use of smartphones is preferable against the use of electronic whiteboard icons to communicate laboratory test results, as otherwise, this forces physicians and nurses to primarily access whiteboard information on permanent static screens also having in mind that physicians do not pay enough attention to the icons [41,42].

Real-time push notifications on mobile apps can significantly improve patient care by transforming the paradigm from the one that requires caregivers to search for information in the EHR among hundreds of data elements to the one where specific data are actively presented to frontline caregivers for timely decision making [43]. This could make a major contribution to speeding up emergency care provision and ultimately shortening the total ED-LOS whether for discharge or hospital admission. For example, push notifications of a rapid influenza test to ED physicians reduced the time to cancellation of an isolation order, the time to transfer to an inpatient unit, and the ED-LOS by approximately one hour among ED patients presenting with suspected influenza [43]. An extensive number of publications have also examined the impact of alert notifications on critical laboratory results. Although sometimes contradictory or pointing out the risk of alert fatigue potentially slowing the response to these alerts and harmful to the patient [43,44,45], they mostly show a significant reduction in time lag between laboratory result availability and decision making, a high degree of clinician approval, and a beneficial impact on patient care [25,46,47,48,49]. During 2018, there were an estimated 130 million ED visits in the USA. Two-thirds of patients spent more than two hours in the ED, including 34.8% between two and four hours and 29.1% with an extended ED-LOS greater than 4 h [32]. Reducing the time to interpretation of results could potentially help reduce these delays. In addition to saving time, such an app likely streamlines caregiver mobility by reducing incessant travel within the ED between the caregiver’s current location and their workstation. However, this hypothesis has not yet been verified in the present study and will be the subject of the forthcoming time-motion trial.

Working in EDs requires collaboration between healthcare workers from different professions in the delivery of patient care, with frequent interaction between staff members to communicate patient and related information [50]. However, the contextual complexity of the high-risk, time-constrained ED environment segregates caregivers from one another and presents a challenge to optimal interprofessional information exchange that is often interrupted and fragmented [11,51,52,53]. Poor communication wastes time and is associated with inefficient patient care, loss of information, jeopardized patient safety, and lower job satisfaction [12,13,54]. The U.S. Joint Commission International has made interprofessional communication one of its main goals to improve patient safety [55]. Information and communication technologies could contribute to this goal. However, few studies have evaluated the benefits of using apps on personal mobile devices to improving communication among caregivers in the ED setting. Among these, some authors have reported the use of the popular WhatsApp app (Meta Platforms Inc., formerly known as Facebook, Menlo Park, CA, USA) [56,57], but this has raised security concerns about privacy compromises [56,58]. Moreover, WhatsApp is not primarily targeted at caregivers and was not developed in a user-centered way for this purpose, especially not for EDs and their constraints. Kentab et al. recently reported good effectiveness of using a customized, smartphone-based, push-to-talk app among ED workers for sharing instant voice messages during the COVID-19 outbreak to help optimize staff infection-control measures [59], but ED-LOS was not measured. Studies have also reported improved communication using two-way text messaging over pagers but, again, without focusing on its impact on ED-LOS [60,61]. Our preliminary findings thus add knowledge to the limited number of published studies examining the contribution of mobile apps to improve communication within an ED. This could help open up interesting prospects to reduce interprofessional communication disruptions that emergency medicine often struggles to solve with expensive and ill-suited means.

## 5. Limitations

This pilot study has some limitations. First, the sample size was small although it anticipated a large effect that was reflected in the results. Second, we did not measure the distance caregivers walked with or without the app. This will be one of the outcomes of our future trial based on a digital pedometer. Third, the quality of information exchanged during interpersonal communication was not assessed. This will also be the subject of the future larger-scale trial. Fourth, this study was conducted in a semi-simulated environment to limit potential confounders and standardize the intervention. Indeed, high-fidelity simulation is an essential method to evaluate research questions and technologies when the chaotic environment, where the intervention takes place, complicates the task [62]. Fifth, although not evaluated in this study, it might be of interest in the main study to determine whether specific alerts, such as a flashing icon or sound for flagged pathological values of laboratory findings, have an impact on outcomes. Finally, we did not weight the results obtained with respect to ED crowding between participants. Although the number of participants was modest, it is assumed that the randomized design of the study balanced this variable between the two groups, especially since the study was conducted on a single day, limiting the risk of weekly or seasonal variation. This point will be taken into account in the upcoming main study.

## 6. Conclusions

The use of a mobile app designed to automatically connect ED physicians and nurses to each other in order to collaboratively manage patients and remotely access real-time laboratory results on the move showed a reduction in caregiver turnaround times for consideration of laboratory results and a trend toward reduced time to find colleagues. Dedicated mobile apps have the potential to improve the efficiency of information sharing and emergency care by virtually reducing the distance between ED caregivers and their workstations or co-workers in a simple, lightweight, and replicable manner. This addresses one of the key ED-LOS issues, i.e., the time professionals waste searching for information and wandering around for colleagues, rather than having to painstakingly alter existing facility spatial organization and architecture. The results generated by this semi-simulated study will inform the conduct of a larger randomized, controlled trial.

## Figures and Tables

**Figure 1 jpm-12-00428-f001:**
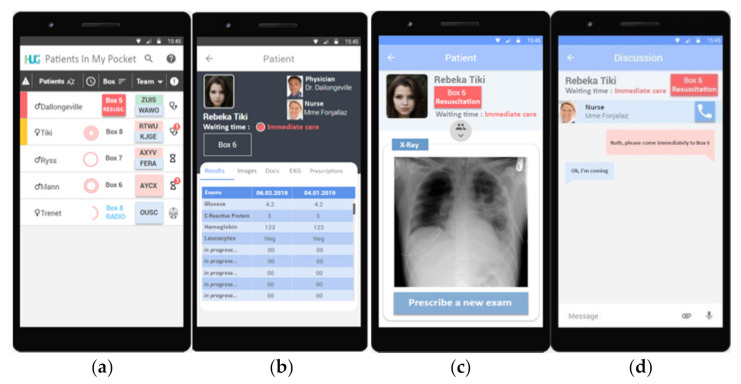
PIMPmyHospital screenshots. (**a**) Color-coded bars represent patient status on the five-level Canadian Triage and Acuity Scale [31]; patient gender and identity; time since admission as radial circular timers; patient allocation per room; color-coded box of individuals in charge of the patient; and patient status (seen by a physician, waiting for results, CT-scan in progress, etc.). Push notifications prompt the user to be aware of the situations. (**b**,**c**) Selecting a patient opens a new page with scrollable contextual tab menus containing information related to laboratory results, imaging, patient files, electrocardiograms, and prescriptions entered in the computerized institutional prescription software. (**d**) The instant messaging system. The logos at the bottom of the screen represent the possibility to also import and link documents to the conversion as well as to send voice memos. Adapted from [26].

**Figure 2 jpm-12-00428-f002:**
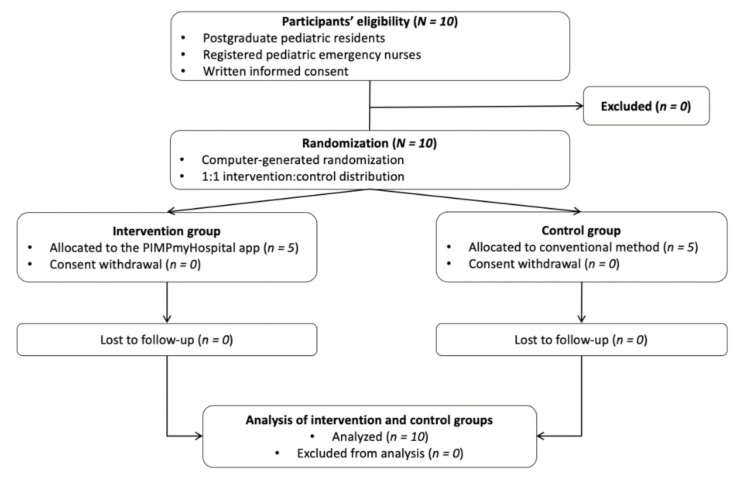
CONSORT flowchart.

**Figure 3 jpm-12-00428-f003:**
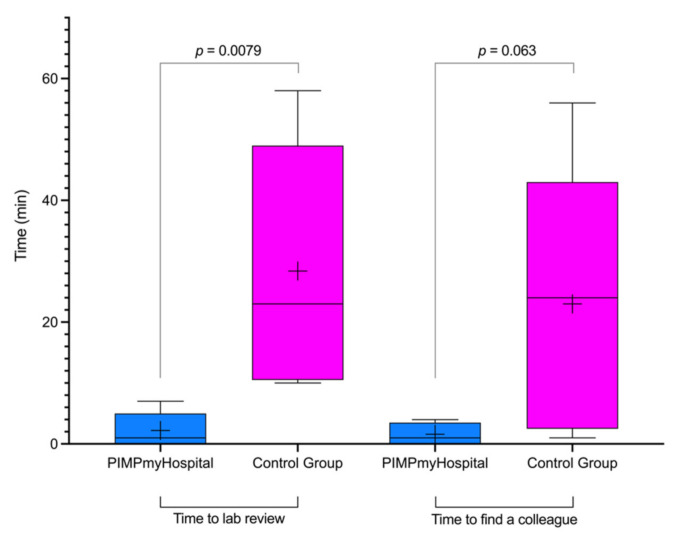
Boxplots of time to laboratory review and time to find a colleague when using the mobile app compared to conventional methods. Solid horizontal lines denote median and IQR; the endpoints of the whiskers indicate the range. The long upper whiskers show that participants were more varied among the most positive quartile groups. The cross denotes the mean.

**Figure 4 jpm-12-00428-f004:**
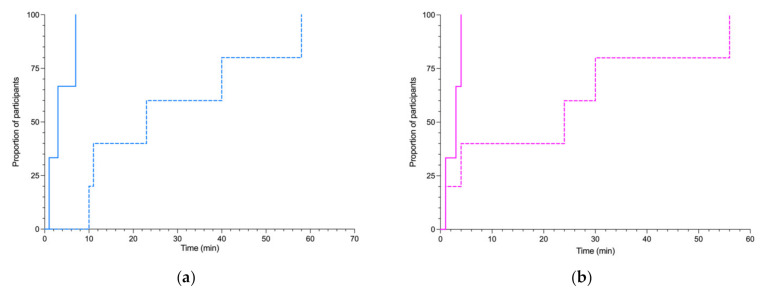
Time to laboratory consideration and to find a colleague. (**a**) Kaplan–Meier curves of time elapsed from the laboratory report to consideration of results and (**b**) time from request to find a colleague to a physical connection using the PIMPmyHospital app (plain lines) vs conventional methods (dashed lines). Log-rank (Mantel–Cox) test comparing curves: χ^2^ = 8.2 and *p* = 0.004 for laboratory results; χ^2^ = 2.7 and *p* = 0.1 for finding a colleague.

**Table 1 jpm-12-00428-t001:** Baseline characteristics of participants.

	PIMPmyHospital (*N* = 5) ^a^	Control Group (*N* = 5)
Age in years, mean (SD)	34.2 (8.5)	30.8 (3.7)
Age in years, *n* (%)		
<30	1 (20.0)	2 (40.0)
30−39	3 (60.0)	3 (60.0)
≥40	1 (20.0)	0 (0)
Gender, *n* (%)		
Female	4 (40)	4 (40)
Male	1 (10)	1 (10)
Work experience in years since certification, mean (SD)	10.8 (9.1)	5.8 (3.4)
Work experience in years since certification, *n* (%)		
<5	1 (20.0)	2 (40.0)
5−9	2 (40.0)	2 (40.0)
≥9	2 (40.0)	1 (20.0)
Work experience in months in the PED, mean (SD)		36.6 (26.6)
Work experience in months in the PED, *n* (%)		
<12	1 (20.0)	1 (20.0)
12−24	1 (20.0)	1 (20.0)
≥24	3 (60.0)	3 (60.0)
Satisfaction ^b^ with current timelines from laboratory report to review, mean (SD)		
	4.4 (1.1)	5.0 (3.0)
Satisfaction ^b^ with current situation to find a peer, mean (SD)		
	3.6 (0.5)	2.2 (1.8)

^a^ N, number of participants; ^b^ on a 10-point Likert scale (0 = not at all satisfied, 10 = extremely satisfied).

## Data Availability

The data presented in this study are available in deidentified form on request from the corresponding author. The data are not publicly available due to privacy restrictions.

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
