# Peer review of "Effectiveness of a Mobile App in Reducing Therapeutic Turnaround Time and Facilitating Communication between Caregivers in a Pediatric Emergency Department: A Randomized Controlled Pilot Trial"

_jpm, 2022, doi:10.3390/jpm12030428_

Round 1

Reviewer 1 Report

This is a well written and interesting study examining an important topic in the ED.  The introduction, methods and results are fine.  The discussion, although well written is somewhat long.  One area that the authors do not discuss is HIPAA and patient privacy-if providers are using their own mobile devices, how is medical data protected? 

Reviewer 2 Report

Authors correctly recognized the importance of small volume of participants, ant they wrote this paper is a pilot-study and requires further investigation.

Maybe, for future investigation, one could measure the similar outcomes if pathological values of laboratory findings will be flagged by blinking on the screen or by sound.
